# Effect of *Paulownia* Leaves Extract Levels on In Vitro Ruminal Fermentation, Microbial Population, Methane Production, and Fatty Acid Biohydrogenation

**DOI:** 10.3390/molecules27134288

**Published:** 2022-07-03

**Authors:** Bogumiła Nowak, Barbara Moniuszko-Szajwaj, Maria Skorupka, Julia Puchalska, Martyna Kozłowska, Jan Bocianowski, Paweł Antoni Kołodziejski, Małgorzata Szumacher-Strabel, Amlan Kumar Patra, Anna Stochmal, Adam Cieslak

**Affiliations:** 1Department of Animal Nutrition, Poznan University of Life Sciences, Wolynska 33, 60-637 Poznan, Poland; bogusianowak07@wp.pl (B.N.); skorupka.maria@wp.pl (M.S.); puchalska00@icloud.com (J.P.); martyna.kozlowska@up.poznan.pl (M.K.); malgorzata.szumacher@up.poznan.pl (M.S.-S.); 2Department of Biochemistry, Institute of Soil Science and Plant Cultivation, State Research Institute, Czartoryskich 8, 24-100 Pulawy, Poland; bszajwaj@iung.pulawy.pl (B.M.-S.); asf@iung.pulawy.pl (A.S.); 3Department of Mathematical and Statistical Methods, Poznan University of Life Sciences, Wojska Polskiego 28, 60-637 Poznan, Poland; jan.bocianowski@up.poznan.pl; 4Department of Animal Physiology, Biochemistry and Biostructure, Poznan University of Life Sciences, Wolynska 35, 60-637 Poznan, Poland; pawel.kolodziejski@up.poznan.pl; 5Department of Animal Nutrition, West Bengal University of Animal and Fishery Sciences, 37 K. B. Sarani, Kolkata 700037, India; patra_amlan@yahoo.com

**Keywords:** feed alternatives, methane production, ruminal microbe, ruminal fermentation

## Abstract

Paulownia is a fast-growing tree that produces a huge mass of leaves as waste that can be used as a feed source for ruminants. The previous study showed that phenolic compounds were the most active biological substances in *Paulownia* leaves, which affected the ruminal parameters and methane concentration. However, there are no scientific reports on the *Paulownia* leaves extract (PLE) containing phenolic compounds for their mode of action in the rumen. Phenolics constituted the main group of bioactive compounds in PLE (84.4 mg/g dry matter). PLE lowered the concentration of ammonia, modulated the VFA profile in the ruminal fluid, and decreased methane production. The PLE caused a significant reduction of in vitro dry matter degradability, reduced the number of methanogens and protozoa, and affected selected bacteria populations. PLE had a promising effect on the fatty acid profile in the ruminal fluid. *Paulownia* as a new dietary component or its extract as a feed additive may be used to mitigate ruminal methanogenesis, resulting in environmental protection and reducing ruminal biohydrogenation, improving milk and meat quality.

## 1. Introduction

Growing consumer awareness [1] and the need to feed a growing population and protect the natural environment [2] are the driving forces behind the search for alternative feed components that do not compete with human foods, and are an ally in climate protection. These include alternative dietary components used in cattle nutrition [3] and waste management. Ecologists focus on implementing those solutions in high-tech and urbanized countries, where such adverse impacts are evident [1]. Agriculture is an inherent source of greenhouse gases, which is assumed to be responsible for 10–12% of total greenhouse gases caused by human activity [4], of which half of those emissions is produced by ruminant livestock production [5]. Moreover, with the increasing demand for animal products, ruminants could contribute more to global methane emissions unless methane mitigation measures are not adopted [6,7].

Different nutritional strategies are implemented in livestock production to mitigate methane emissions. Recently, many biologically active substances and tree leaves have been used as a source of nutrients and plant bioactive compounds to decrease methane production and improve ruminal fermentation [3,8,9]. An example could be a *Paulownia* tree [10]. *Paulownia* dried leaves or silage positively affected the rumen fermentation parameters by changing acetate to a propionate ratio without any adverse effect on nutrient digestion, which could be exploited in the nutrition of ruminants [3,9]. It has also been shown that phenolic compounds, including phenolic acids and flavonoids, are the most biologically active substances in *Paulownia* leaves. These compounds can influence the rumen fermentation processes, including methanogenesis. The study carried out by Puchalska et al. [3] showed that *Paulownia* leaves containing phenolic compounds were perhaps responsible for reducing the methanogenesis process by directly inhibiting the population of methanogens. In the study with *Paulownia* leaves or silage, the confounding effects of nutrients may exist on ruminal fermentation and methanogenesis. Using a *Paulownia* leaves extract (PLE) containing a high concentration of plant bioactive compounds, including phenolics, may avoid the influence of nutrients on microbial fermentation. However, there are no scientific reports on the mode of action of PLE in the rumen. Thus, we hypothesized that secondary metabolites present in high concentrations in PLE could alter the rumen microorganism populations, thus, modulating ruminal fermentation and mitigating methane emission. The present study aimed to explore the influence of biologically active substances contained in the PLE on in vitro rumen fermentation, microbial populations, and the fatty acid profile in buffered rumen fluid.

## 2. Results

### 2.1. Concentration of Active Compounds in the Extract

The UHPLC-DAD-ESI-MS results of PLE are shown in Table 1. The main bioactive compounds in PLE were phenolic compounds (84.4 ± 2.50 mg/g DM), mainly verbascoside, and their derivatives, such as methoxyverbascoside, hydroxyverbascoside I, and hydroxyverbascoside I (29.0 mg/g DM). Moreover, iridoids, namely, 7-hydroxytomentoside or aucubin (14.9 ± 0.80 mg/g DM) and very trace amounts of catalpol, were present. The extract also contained small amounts of triterpenoids (2.59 ± 0.133 mg/g DM). Maslinic acid and its equivalents, C_30_H_48_O_3,_ C_30_H_48_O_4,_ and C_30_H_48_O_5_, were present in the PLE extract.

### 2.2. Rumen Fermentation, Gas Production, and Microbial Populations

A decrease in pH (linear decrease, *p* = 0.05) was observed with increasing doses of PLE, with the highest pH at 100 mg/kg of PLE (Table 2). Ruminal ammonia concentration decreased linearly, quadratically, and cubically (*p* < 0.001), with increasing concentrations of PLE with the lowest ammonia concentration at the highest doses. PLE inclusion also affected the total VFA concentration (*p* < 0.001). The inclusion of PLE linearly, quadratically, and cubically (*p* < 0.001) increased the molar proportion of propionic, isobutyric, and isovaleric acids. However, the molar proportion of acetic acid decreased linearly, quadratically, and cubically (*p* < 0.001). The acetate to propionate ratio decreased linearly, quadratically, and cubically (*p* < 0.001). Molar proportions of butyric and valeric acids were not affected by PLE. IVDMD was affected by the PLE in a linear, quadratic, and cubic manner (*p* < 0.001). Logarithmic contrasts were significant for all parameters except CH_4_ in mmol/g DM, CH_4_ in mmol/L total gas, and *Isotrichidae* in 10^3^/mL (Table 2).

With increasing PLE doses, the total gas and methane production (mmol/g DM, mmol/L total gas or mmol/g IVDDM) decreased linearly, quadratically, and cubically (*p* < 0.001). The inclusion of PLE resulted in a linear, quadratic, and cubic decrease of *Ophyroscolecidae* and total protozoa (*p* < 0.001), but *Isotrichidae* was not affected (*p* > 0.10). Due to the PLE inclusion, methanogens and total bacteria populations decreased linearly, quadratically, and cubically (*p* < 0.001).

### 2.3. Bacteria Quantification

The inclusion of PLE affected the microbial populations. *Fibrobacter succinogenes, Ruminococcus albus*, *Butyrivibrio fibrisolvens*, and *Ruminococcus flavefaciens* populations decreased linearly, quadratically, and cubically (*p* < 0.001) with increasing PLE concentrations (Table 3). *Anaerovibrio lipolytica*, *Butyrivibrio proteoclasticus*, *Prevotella* spp. *Megasphaera elsdenii*, and *Streptococcus bovis* populations increased linearly, quadratically, and cubically (*p* < 0.001) with increasing PLE concentrations. Logarithmic contrasts were significant for all items, except *Anaerovibrio lipolytica*, *Megasphaera elsdenii*, and *Streptococcus bovis* (Table 3).

### 2.4. Fatty Acids Profile

With increasing PLE doses, C18:1 trans-10, C18:2 cis-9, cis-12, C18:3 cis-9, cis-12, cis-15 (3n3), C18:2 trans-10, cis-12, the sum of UFA, the sum of MUFA, the sum of PUFA, the sum of n−6 FA, the sum of n-3 FA, the sum of n6 PUFA, the sum of n3 PUFA, PUFA/SFA ratio, the sum of C18:1, the sum of trans C18:1, the sum of LCFA, and DI C18:1 cis 9/(C18:0+C18:1 cis 9) increased in a logarithmic manner (*p* ≤ 0.05; Table 4). The rest of the fatty acids decreased logarithmically (*p* ≤ 0.05), with the exceptions of C14:0, C16:0, C18:1 cis-9, the n-6/n-3 FA ratio, the LNA/LA ratio, the sum of cis C18:1, DI, and DI RA/(VA+RA). Significant linear decreases were observed only for C18:3n-6 and the sum of other FA; however, a significant cubic effect for the sum of trans C18:1 (*p* ≤ 0.05) was noted with the highest concentration at 400 mg/kg of PLE.

## 3. Discussion

The present study demonstrated the effects of the use of PLE on the in vitro ruminal fermentation parameters, including the fatty acids profile, ammonia concentration, pH, gas production, and microbial populations. To our current knowledge, the experiment examining the influence of PLE containing plant bioactive compounds on ruminal in vitro fermentation and microbial populations has not yet been carried out. Previous studies evaluated the effect of *Paulownia* leaves in ruminant diets as the silage or dried leaves [3,9,11], which can exert confounding effects along with the plant bioactive compounds present in *Paulownia* leaves.

The PLE contained mainly phenolic compounds (84.4 mg/g DM). Verbascoside (acteoside) and its derivatives were the principal compounds in this group (total 29.0 mg/g DM). Other biologically active substances such as iridoids (e.g., 7-hydroxytomentoside and aucubin) and triterpenoids (e.g., maslinic acid and equivalents of maslinic acid) were also present. Phenolic compounds, including derivatives of verbascoside are widespread compounds present in higher plants, especially in the *Araliaceae*, *Bignoniaceae*, *Crassulaceae*, *Labiateae*, *Oleaceae*, *Plantaginaceae, Polygonaceae*, *Scrophulariaceae*, *Smilaeaceae*, and *Verbenaceae* plant families [12,13]. The previous studies reported that verbascoside in *Paulownia* leaves ranged from 1.7 to 21.0 mg/g DM and phenolic compounds ranged between 2.53 to 33.0 mg/g DM [3]. Navarette et al. [14] reported that plantain (*Plantago lanceolata* L.) and chicory (*Cichorium intybus* L.) contain verbascoside in the range of 0.5 to 42 mg/g DM. The discrepancies are noted due to the differences in the time of harvest, agronomic management, soli condition, and variety variance. The activity of verbascoside as well as its derivatives was mainly assessed for anti-inflammatory and antioxidant activity [15,16,17,18]. However, research was also conducted on verbascoside in terms of changes in the ruminal environment, as well as milk and meat quality of the cattle [19,20]. Studies on verbascoside and its derivatives as one of the major groups of phenolic compounds in *Paulownia* leaves, with the content of verbascoside at 21 and 10 mg/g DM in dried and silage forms, respectively, were carried out [3,9]. However, up until now, no research on the *Paulownia* extract containing significant concentrations of phenolic compounds had been conducted. The investigation of the PLE on the ruminal in vitro parameters could lead to the possibility of using *Paulownia* as a new dietary component or PLE as a feed additive for potentially modulating ruminal methanogenesis (environmental protection) and biohydrogenation (improvement of milk and meat quality).

The results confirmed the positive effect of PLE containing a high concentration of derivatives of verbascoside and verbascoside itself on the basic ruminal parameters. PLE reduced the ammonia concentration, modulated the VFA and FA profile, and decreased methane production, the number of methanogens, and IVDMD. In the present study, the amount of dietary verbascoside derivatives was 29.0 mg/g DM. In contrast, in a previous experiment where dried leaves were used, the highest verbascoside concentration was 21 mg/g DM, which did not affect the IVDMD [3]. The results suggest that nutrient components (e.g., crude protein) may interact with biologically active substances or that biologically active substances may become physically less accessible to the microflora, resulting in reduced microbial activity in experiments using *Paulownia* leaves relative to PLE. Navarette et al. [14] tested the commercial form of verbascoside (99% pure; Extrasynthese S.A, France) at 36 or 40 mg/g DM, depending on the source of the substrates, i.e., plantain or chicory and the majority of ruminal parameters were determined, except IVDMD. There was no change in gas production, which is direct evidence of the lack of an IVDMD reduction. The authors emphasized the improved ruminal fermentation parameters in the verbascoside supplemented group. Getachew et al. [21] indicated the possibility of using verbascoside as an energy source due to the non-specific glycosidase. In the present research on PLE, no such positive changes were noted. This could be due to the lower amount of verbascoside compared to its derivatives in the extract. Similarly, in the study by Özelçam et al. [11], IVDMD decreased with increased polyphenolic compounds in the *Paulownia* materials. We assumed that the interaction of verbascoside derivatives or other biologically active substances contained in PLE caused a reduction of IVDMD.

Linear decreases in pH value (*p* = 0.05) were observed with the increasing dietary concentration of the extract, though the total VFA concentration decreased linearly (Table 2). This reduction is consistent with the results obtained in a batch culture study on a saponin-rich extract [22]. In earlier studies, the experimental groups with dried *Paulownia* leaves or ensiled *Paulownia* were rich in secondary metabolites, but at lower concentrations, which ranged from 2 to 21 of dry weight vs. 48% of dry weight detected in the current study. The modulating effects of the extract on the population of microorganisms, which resulted in changes in the fermentation processes by altering the functionality of ruminal microorganisms, could result in changes in the pH values [23,24]. In the study by Arhab et al. [25], a comparison among different solvents was made (methanol, ethanol, acetone), and no effect on the pH was found. In addition, ethanol in our experimental groups did not change the pH value; therefore, it was decided not to include an additional blank group column in Table 2. Additionally, a decrease in ammonia concentration (*p* < 0.001) was observed with an increasing PLE concentration. This may result from a reduction in the number of protozoa [26]. The direct inhibition of microorganisms’ activity by using verbascoside or aucubin in experimental groups is indicated as the cause of these changes [14]. The same properties might be exhibited by the verbascoside derivatives in PLE. Additionally, the reduction in ammonia concentration may result from the ability of polyphenolic compounds to form complexes with proteins derived from the incubated substrate and may partially inactivate proteolytic enzymes [27]. As a consequence, proteins are not available for ruminal microorganisms. The complexes may directly reduce the concentration of ammonia in the ruminal fluid. Furthermore, the number and activity of proteolytic bacteria such as *Butyrivibrio fibrisolvens*, which was confirmed in our in vitro study (*p* < 0.001), may be responsible for lowering the ammonia concentration. Nevertheless, the activity of *Prevotella* spp. increased (*p* < 0.001); therefore, it seems that their activity increases when the population of cellulolytic bacteria in the ruminal fluid with a lower pH decreases since they are considered to be one of the main starch-degrading bacteria [28].

The observed changes in the number of individual bacteria may explain the alterations in the VFA profile. The VFA analysis confirmed the reduction of the acetic acid content, which is consistent with the reports of Klevenhusen [26] and Cardozo [29] studying plant bioactive compounds. This may be related to the pH decline in experimental trials, which is related to both the sensitivity of cellulolytic bacteria to low pH values and the presence of fibrolytic bacteria as the main producer of rumen acetic acid [30,31]. In addition, a low pH shifts the fermentation towards lactic acid by *Streptococcus bovis* instead of acetic acid, because of pyruvate formate lyase inhibition [32]. It increases *Megasphaeraelsdenii* population and prevents lactic acidosis by using lactic acid that is harmful to the other pH-sensitive bacteria [33]. However, the accumulation of lactic acid in the ruminal fluid occurs significantly when the pH is below 5.5 [34]. We observed a reduction in the population of *Fibrobacter succinogenes*, *Ruminococcus albus,* and *Ruminococcus flavefaciens*, and an increase in the population of *Megasphaera elsdenii* and *Streptococcus bovis*. The bacterial growth observed in the present study may cause the reduction of competition among microbial populations. This is consistent with studies conducted with dietary essential oils [35]. The study by Hobson and Stewart [36] showed that *Fibrobacter succinogenes* is sensitive to plant secondary metabolites, including phenolic monomers, which is consistent with the results of the present experiment. The *Ruminococcus flavefaciens* population was also reduced.

It should be noted that despite a significant increase in the population of some types of bacteria, the total protozoa and methanogens counts in the ruminal fluid were significantly reduced by PLE. Moreover, the results mentioned above confirmed that polyphenolic compounds reduced methane production through a negative impact on the activity of the population of the methanogens [37]. This is due to the inhibition of the growth of methanogens, which is directly related to the protozoa population (protozoa-associated methanogens, PAMs). PAMs are responsible for 37% of total methane production [38]. Plant phenolic compounds can also directly inhibit the growth of methanogens [39]. Changes in the VFA profile in favor of an increased concentration of propionic acid and butyric acid may also cause a hydrogen shift from the methane pathway, making it available for propionic acid production [40].

PLE containing phenolic compounds as the main bioactive group, including verbascoside derivatives, has a promising effect on the ruminal fluid’s fatty acid profile, which may improve the milk’s fatty acid profile. An increase in UFA, including MUFA and PUFA, was observed in the experimental group containing the highest concentration of PLE. A decrease in the concentration of C15:0 was noted, which may be associated with a reduction of the *Fibrobacter succinogenes* population as this bacterium is responsible for the synthesis of some FAs, including odd-chain FAs (mainly C15:0) [41]. A higher concentration of C18:1 trans-10, C18:2 cis-9 cis-12, and a lower concentration of C18:2 cis-9 trans-11 may suggest the influence of one of the verbascoside derivatives and other bioactive compounds contained in *Paulownia* on the inhibition of the biohydrogenation process by limiting the activity of group A bacteria, mainly *Ruminococcus albus* and *Butyrivibrio fibrisolvens* [42]. It is likely that an increase in C18:2 trans-10 cis-12 may be due to the increased activity of *Megasphaera elsdenii* [43]. With the addition of *Olea europaea L.* fruit pomace, rich in polyphenols, the content of C18:2 cis-9 cis-12 and C18:3 cis-9 cis-12 cis -15 in the ruminal fluid was higher than in the control, which was in line with our results [44].

## 4. Materials and Methods

All procedures were performed following the guidelines of the National Ethical Commission for Animal Research (Ministry of Science and Higher Education, Warsaw, Poland). The study was approved by the Local Ethical Commission (permission no. 14/2019).

### 4.1. Plant Material and Preparation of the Extract

Fresh *Paulownia* Clone in Vitro 112 leaves were procured from a plantation nearby Trzcianka, Greater Poland Voivodeship, Poland. Then, the leaves were chopped into smaller pieces of 1.5 cm size, frozen and freeze-dried (Gamma 2–16 LSC, Christ, Osterode, Germany), ground using a ZM 200 mill (Retsch, Düsseldorf, Germany; 1 mm sieve), and kept in the dark for further use and analysis.

The extract of paulownia leaves was prepared in sequential extractions. The ground plant material (100 g) was extracted with 1.5 L of 5% methanol (*v*/*v*) using an ultrasonic bath at room temperature for 30 min, and also macerated using a magnetic stirrer for 30 min at room temperature. The content was centrifuged at 4000× *g* for 10 min. The residue was re-extracted with 1.5 L of 30% methanol (*v*/*v*) under the same conditions as above and centrifuged at 4000× *g* for 10 min. Finally, the residue was re-extracted with 1.5 L of 70% methanol and centrifuged at 4000× *g* for 10 min. The supernatants were pooled, concentrated under reduced pressure, freeze-dried, and designated as paulownia leaves extract (PLE). The yield of the extraction was 38.02%. The plant material was extracted in 8 runs with 100 g in each run.

### 4.2. Phytochemical Analysis of Paulownia Leaves Extract

The phytochemical composition of the extract from leaves of *Paulownia* Clone in Vitro 112 was determined by UHPLC-DAD-ESI-MS and performed in triplicate. Chromatographic separations of the samples were performed using an Acquity UPLC system (Waters, Milford, MA, USA), coupled with an Acquity TQD (Waters) mass detector on an Acquity BEH C18 column (100 mm × 2.1 mm, 1.7 µm; Waters). The mobile phase was composed of mixtures of solvent A (0.1% formic acid (FA) in Milli-Q water) and solvent B (0.1% FA in acetonitrile). The constituents of the samples were identified based on their molecular mass (MS), ultraviolet (UV) spectra (identification was supported by spectra and molecular formulas obtained during previously performed LC-HRMS/MS analysis (Q-TOF) of a similar extract from *Paulownia* Clone in Vitro 112 leaves; data not shown), and literature data. Phenolic compounds and triterpenoids concentrations were determined according to previously used procedures [17]. In the case of iridoids, a minor modification of the previous chromatographic method [17] was applied, and the following elution program was used: 0–2.5 min: 2% B; 2.50–10.0 min: 2-40% B; 10.10–12.10 min: 99% B; 12.20–16 min: 2% B. Phenolic compounds were semi-quantified based on UV chromatograms (λ = 330 nm). Concentrations of individual phenolics were determined using calibration curves of verbascoside (HWI Analytik, Rüelzheim, Germany) and rutin (PhytoLab, Vestenbergsgreuth, Germany). The contents of the phenylethanoids, as well as other derivatives of phenolic acids and all minor and unidentified compounds, were expressed as an equivalent of verbascoside, and the contents of the major flavonoids were expressed as a rutin equivalent.

Iridoid concentrations were semi-quantified by LC-MS, using a negative ion SIM method. Two ions were monitored: the formic acid adduct ion of catalpol (*m*/*z* 407), formic acid adduct ion of 7-hydroxytomentoside/aucubin (*m*/*z* 391). The iridoid content was determined on the basis of a calibration curve of catalpol (Sigma, St. Louis, MO, USA), and expressed as an equivalent of catalpol.

Triterpenoid concentrations were determined by LC-MS, using a negative ion SIM method. The following ions were monitored (in sequence, within set time ranges): *m*/*z* 503, *m*/*z* 487, *m*/*z* 471, and *m*/*z* 455 (chosen on the basis of our preliminary qualitative analyses). The triterpenoid content was determined on the basis of a calibration curve of maslinic acid (Sigma), and expressed as an equivalent of maslinic acid.

### 4.3. Experimental Design and Treatments

The experiment was performed using an in vitro short-term batch culture fermentation method. The in vitro experiment consisted of a control group, the ethanol group (blank group), and six experimental groups (six doses of PLE) with three repetitions in each group (total eight groups × three bottles), and was completed in three consecutive runs (on different days) with 24 h of incubation in each run. The experimental groups were supplemented with PLE at six doses (100, 200, 400, 600, 800, and 1000 mg of PLE/L of buffered ruminal fluid). All six doses (representing 4, 8, 16, 24, 32, and 40 mg per bottle) were dissolved in 100 µL of ethanol and added to the substrate 2 h before the start of filling the bottle with buffered rumen fluid. A small amount of ethanol addition 2 h before the start of the experiment allowed evaporation of ethanol from the substrate.

### 4.4. In Vitro Batch Culture

Ruminal fluid from three ruminal cannulated Polish Holstein-Friesian dairy cows was collected for in vitro ruminal inocula. The average weight of the animals was 625 ± 25 kg body weight. They were fed approximately 23 kg of dry matter (DM) per day. The diet consisted of 76:24 forage to concentrate and included maize silage (388 g/kg DM), alfalfa silage (82 g/kg DM), grass silage (91 g/kg DM), beet pulp (103 g/kg DM), brewer’s grain (95 g/kg DM), extracted rapeseed meal (108 g/kg DM), a concentrate mixture (119 g/kg DM), and mineral–vitamin mixture (14 g/kg DM). Water was available *ad libitum*. The rumen fluid was collected before the morning feeding from each cow. Rumen fluid was squeezed through two layers of cheesecloth into a bottle (Schott North America Inc., Elmsford, NY, USA), pooled, and finally mixed in an equal ratio. Then, ruminal fluid was transported to the laboratory (in anaerobic conditions at 39 °C) within 30 min and used as a source of inoculum.

The procedure for preparing batch culture was carried out following the method described by Cieslak et al. [45]. Briefly, the rumen fluid was mixed with the buffer solution (292 mg dipotassium hydrogen phosphate, 240 mg potassium dihydrogen phosphate, 480 mg ammonium sulfate, 480 mg sodium chloride, 100 mg magnesium sulfate heptahydrate, 64 mg calcium chloride dihydrate, 4 mg sodium carbonate anhydrous, and 600 mg/L cysteine hydrochloride in 1 L of distilled water) at a 1:4 (*v/v*) ratio. The bottles were constantly purged with O_2_-free CO_2_. Incubation of each experiment was run at 39 °C under CO_2_ in 40 mL buffered rumen fluid added to pre-warmed 125 mL vessels where 400 mg of the substrate was. The basal substrate (the diet of cannulated cows) was supplemented with various doses of PLE. The incubation flasks were sealed with rubber stoppers and aluminum caps, placed in an incubator maintained at 39 °C for 24 h, and mixed periodically.

### 4.5. Sampling and Analysis

The microbial fermentation was stopped after 24 h of incubation. The in vitro ruminal fluid samples were collected for analyses of pH, ammonia concentration, volatile fatty acids (VFA), protozoa, bacteria, and methanogen counts. Furthermore, individual bacteria quantification and fatty acids (FA) profile analyses in ruminal fluid samples were performed. The rumen fluid samples were stored at −20 °C for further analysis regarding FA composition. Subsequently, two samples of each group of filtered rumen fluid (3.6 mL) were preserved with 0.4 mL of 46 mM HgCl_2_ solution and stored at −20 °C for determination of VFA and NH_3_. Another rumen fluid sample was strained through two layers of cheesecloth and the rumen fluid was fixed with 4% formalin (6:1 ratio) for protozoa counting. For bacteria and methanogen analysis, samples were frozen in liquid nitrogen directly after collection, and then transferred to a −80 °C freezer and kept there until analysis.

#### 4.5.1. Ruminal Fermentation, Methane Production, Microbial Population, and Fatty Acids Profile

After 24 h of incubation, the gas production was measured by the displacement of the syringe piston that was connected to the incubation flask. The net gas production was calculated by subtracting gas production in blank flasks (without a substrate), from the total gas produced in the flasks consisting of the buffered rumen fluid and substrate. Gas samples (500 μL) were collected from the headspace of flasks in a gas-tight syringe (GASTIGHT^®^ Syringes, Hamilton Bonaduz AG, Bonaduz, Switzerland) for methane analysis. The sampled gas was injected into an SRI 310 gas chromatograph (SRI310, Alltech, PA, USA), equipped with a thermal conductivity detector (TCD) and Carboxen—1000 column (mesh side 60/80, 15 FT × 1·8 INS.S, Supelco, Bellefonte, PA, USA) according to the procedure described by Cieslak et al. [45]. Rumen fluid pH was immediately measured after 24 h of incubation (pH-meter CP-104, Elmetron, Zabrze, Poland). The colorimetric Nessler’s method described by Cieslak et al. [45] was used to determine the ammonia concentration. The VFA concentrations were measured by gas chromatography (GC Varian CP 3380, Sugarland, TX, USA) equipped with a flame ionization detector and a capillary column (30 m × 0.25 mm; Agilent HP-Innowax, 19091N-133, Agilent Technologies, Santa Clara, CA, USA), according to Bryszak et al. [46]. For determination of in vitro dry matter digestibility (IVDMD), the content of incubation flasks was transferred to weighed crucibles. The residues were washed with 50 mL distilled water and dried at 105 °C for 3 days. The IVDMD was calculated as the difference between substrate DM incubated and residue left in the crucible, and was expressed in percentage. The protozoa counts were performed as explained by Bryszak et al. [46], using buffered rumen fluid with a defined volume (100 µL) under a light microscope (Zeiss, type Primo Star no. 5, Jena, Germany). The protozoa were split into *Isotrichidae* and *Ophyosclecidae* groups. Total methanogens and total bacterial populations were quantified by fluorescence in situ hybridization (FISH), according to Puchalska et al. [3]. Determination of the FA content in the ruminal fluid was made with a gas chromatograph (GC Bruker 456-GC, Billerica, MA, USA) equipped with a capillary column (100-m fused silica, 0.25 mm i.d., 0.25 μm film thickness; Chrompack CP7420, Agilent HP, Santa Clara, CA, USA) and a flame ionization detector using 1 μL of sample injected into the column, according to the protocol of Bryszak et al. [46].

#### 4.5.2. Bacterial Quantification

Eight selected species and one genus of rumen bacteria were quantified by real-time PCR. Metagenomic DNA from rumen fluid was extracted using a QIAamp DNA Stool mini kit (Qiagen GmbH, Hilden, Germany), according to Yanza et al. [47]. The specificity of primers was confirmed using the BLAST program in the GenBank Database (Table 5). The quantification of particular bacteria with a known initial DNA concentration (25 ng/µL) was made using the QuantStudio 12 Flex PCR system (Life Technologies, Thermo Fisher Scientific, Waltham, MA, USA). PCR amplification was performed with the Power SYBR Green PCR Master Mix (Thermo Fisher Scientific, Waltham, MA, USA). The mixture for reaction included 4 µL of the 2 × master mix, 25 ng of template DNA, and 0.5 M of each primer in 10 µL of the final volume. The amplification program was used as follows: one cycle of amplification at 95 °C for 10 min for initial denaturation, 45 cycles at 95 °C for 15 s followed by annealing at the temperatures (dependent on the analyzed bacteria) for 5 s, and at the end at 62 °C for 67 s. Products from the fluorescent process were found in the last stage of each cycle. To explain the characteristics of amplification, an analysis of product melting was made after a single amplification (0.1 °C increment from 65 °C to 95 °C with fluorescence collection at 0.1 °C intervals). The relative abundances of DNA copy of each bacterial species vs. total bacteria were calculated using the formula 2^−ΔΔCt^ (RTA).

## 5. Statistical Analysis

The normality of the distribution of all the observed traits was verified with the Shapiro–Wilk’s normality test to check whether the assumption of the residuals’ normal distribution was followed for the analysis of variance (ANOVA). The homogeneity of variance was tested using Bartlett’s test. Box’s M test was used to check multivariate normality and homogeneity of variance–covariance matrices. The arithmetic means and standard error of the mean were calculated. The data were analyzed using one-way ANOVA with the treatment as a fixed factor. Linear, quadratic, cubic, and logarithmic contrasts were tested for all variables. Orthogonal polynomials were used in the analysis of variance for the construction of contrasts among unequally spaced levels of a treatment factor. Contrast is considered significant when the *p*-value is ≤0.05, and tendencies are identified when the *p*-value is >0.05 to ≤0.10. The GenStat v. 18 statistical software package (VSN International) was used for the analyses.

## 6. Conclusions

Secondary metabolites present in the PLE effectively modulated ruminal fermentation and inhibited methane production. The methane production inhibition was dependent on the amount of extract used. The greatest mitigation was noted with the highest dose of PLE in the buffered ruminal fluid (equivalent to 8.4 g phenolic compounds/L of ruminal fluid). This mitigation was directly related to a linear decrease of the degradability of feed (substrate). Therefore, moderate doses of PLE should be used to decrease methane production while not affecting feed degradability to a great extent. The results of the research may constitute the basis for further in vivo research on new phytoadditives in ruminants.

## Figures and Tables

**Table 1 molecules-27-04288-t001:** Contents of the main bioactive compounds (mg/g DM) identified in *Paulownia* leaves extract (PLE).

Compounds	Concentration (mg/g DM)
Phenolic compounds	
Caffeic acid-Hex-dHex	2.7 ± 0.09 *
Luteolin-HexA-HexA	6.5 ± 0.14 ^#^
Hydroxyverbascoside I	5.5 ± 0.12 *
Hydroxyverbascoside II	6.4 ± 0.09 *
Apigenin-HexA-HexA	11.3 ± 0.26 ^#^
Methoxyverbascoside	10.1 ± 0.19 *
Verbascoside	2.6 ± 0.08
Dimethylverbascoside	3.7 ± 0.17 *
Other	35.6 ± 2.11 *
Total	84.4 ± 2.50
Iridoids	
Catalpol	traces
7-Hydroxytomentoside/aucubin	14.9 ± 0.80 ^‡^
Total	14.9 ± 0.80 ^‡^
Triterpenoids	
Total C_30_H_48_O_6_-Hex ^†^	0.45 ± 0.027 ^†^
Total C_30_H_48_O_6_ ^†^	0.25 ± 0.016 ^†^
Total C_30_H_48_O_5_ ^†^	0.92 ± 0.058 ^†^
Maslinic acid	0.34 ± 0.024
Total C_30_H_48_O_4_ ^†^	0.43 ± 0.030 ^†^
Total C_30_H_48_O_3_ ^†^	0.19 ± 0.016 ^†^
Total	2.59 ± 0.133

Hex—hexose; dHex—deoxyhexose; HexA—hexuronic acid. * equivalent of verbascoside; ^#^ equivalent of rutin; ^‡^ equivalent of catalpol; ^†^ equivalent of maslinic acid.

**Table 2 molecules-27-04288-t002:** The effect of *Paulownia* leaves extract (PLE) on in vitro ruminal fermentation and methane (CH_4_) production.

Parameters	CON	PLE mg/kg (DM)						SEM	*p*-ANOVA	Contrast			
		100	200	400	600	800	1000			L	Q	C	Log
Rumen fermentation													
pH	6.33	6.24	6.23	6.20	6.23	6.21	6.22	0.011	0.123	0.05	0.16	0.24	0.008
NH_3,_ mM	19.6	18.6	18.2	18.5	16.3	16.5	16.3	0.198	0.021	<0.001	<0.001	<0.001	0.003
Total VFA, mM	47.3	46.4	46.7	45.8	46.1	46.4	45.1	0.158	0.041	<0.001	<0.001	<0.001	<0.001
Individual VFA, mol/100 mol		VFA, mol/ 100 mol											
Acetate (A)	60.7	55.3	53.9	53.7	51.7	49.8	50.4	0.470	0.002	<0.001	<0.001	<0.001	0.018
Propionate (P)	22.4	25.8	26.5	27.5	27.7	27.8	29.1	0.281	0.273	<0.001	<0.001	<0.001	<0.001
Isobutyrate	1.36	1.28	1.51	1.66	1.57	1.93	1.87	0.046	0.011	<0.001	<0.001	<0.001	0.046
Butyrate	13	15.4	15.4	14.2	16.0	17.0	15.2	0.284	0.578	0.12	0.26	0.42	<0.001
Isovalerate	1.31	0.99	1.45	1.74	1.88	2.20	2.36	0.068	<0.001	<0.001	<0.001	<0.001	0.013
Valerate	1.23	1.23	1.24	1.2	1.15	1.27	1.07	0.035	0.731	0.18	0.16	0.15	<0.001
A/P ratio	2.71	2.14	2.03	1.95	1.87	1.79	1.73	0.038	0.029	<0.001	<0.001	<0.001	<0.001
IVDMD, %	66.8	64.9	62.0	60.0	58.6	56.8	55.3	0.501	0.003	<0.001	<0.001	<0.001	<0.001
		Total gas and methane production											
Total gas, mL/g DM	280	265	264	257	252	235	208	6.57	<0.001	<0.001	<0.001	<0.001	<0.001
CH_4_, mmol/g DM	2.47	2.21	2.13	2.04	1.85	1.73	1.44	0.093	<0.001	<0.001	<0.001	<0.001	0.284
CH_4_, mmol/L total gas	8.34	8.35	8.06	7.93	7.85	7.51	6.62	0.198	0.009	<0.001	<0.001	<0.001	0.096
CH_4_ mmol/g IVDDM	3.70	3.41	3.44	3.40	3.16	3.05	2.60	0.007	0.014	<0.001	<0.001	<0.001	<0.001
		Microbial population											
*Ophyosclecidae*, 10^4^/mL	1.65	1.30	1.20	1.13	1.05	0.93	0.82	0.034	<0.001	<0.001	<0.001	<0.001	<0.001
*Isotrichidae*, 10^3^/mL	2.96	2.77	2.07	1.83	2.30	2.30	2.07	0.111	0.249	0.09	0.23	0.3	0.340
Total protozoa, 10^4^/mL	1.94	1.58	1.41	1.33	1.28	1.16	1.03	0.039	0.027	<0.001	<0.001	<0.001	<0.001
Total bacteria, 10^8^/mL	22.4	17.2	17.9	17.6	16.7	15.7	14.5	0.406	<0.001	<0.001	<0.001	<0.001	<0.001
Total methanogens, 10^7^/mL	5.17	2.86	2.51	2.87	2.06	1.49	1.27	0.151	<0.001	<0.001	<0.001	<0.001	<0.001

CON: control group; SEM: standard error of the mean; Contrast: linear (L), quadratic (Q), cubic (C), and logarithmic (Log) contrasts were estimated for all observed parameters (*p* < 0.05); NH_3_: ammonia concentration; VFA: volatile fatty acids; IVDMD: in vitro dry matter degradability; CH_4_: methane production.

**Table 3 molecules-27-04288-t003:** The effect of *Paulownia* leaves extract (PLE) on in vitro ruminal microbial populations quantified using quantitative PCR.

Items *	CON	PLE (mg/kg DM)						SEM	*p*-ANOVA	Contrast			
		100	200	400	600	800	1000			L	Q	C	Log
*Fibrobacter succinogenes*	3.99	2.40	1.90	2.03	1.73	1.38	1.25	0.184	<0.001	<0.001	<0.001	<0.001	<0.001
*Anaerovibrio lipolytica*	0.97	1.62	1.89	2.76	3.24	4.87	6.56	0.395	<0.001	<0.001	<0.001	<0.001	0.829
*Butyrivibrio fibrisolvens*	1.17	1.38	1.62	1.98	1.24	0.78	0.59	0.095	0.004	<0.001	<0.001	<0.001	0.006
*Butyrivibrio proteoclasticus*	3.25	2.78	6.78	5.68	9.08	12.1	26.2	1.617	<0.001	<0.001	<0.001	<0.001	<0.001
*Prevotella* spp.	4.96	5.66	6.36	7.74	9.81	17.5	22.2	1.330	<0.001	<0.001	<0.001	<0.001	0.038
*Megasphaera elsdenii*	0.54	0.24	0.24	0.51	1.55	1.37	3.16	0.211	<0.001	<0.001	<0.001	<0.001	0.138
*Ruminococcus albus*	0.035	0.027	0.014	0.005	0.004	0.004	0.002	0.003	<0.001	<0.001	<0.001	<0.001	0.039
*Ruminococcus flavefaciens*	0.52	0.23	0.16	0.18	0.08	0.03	0.02	0.034	<0.001	<0.001	<0.001	<0.001	0.016
*Streptococcus bovis*	0.030	0.035	0.036	0.083	0.158	0.289	0.378	0.028	<0.001	<0.001	<0.001	<0.001	0.318

* expressed as an arbitrary unit; Contrast: linear (L), quadratic (Q), cubic (C), and logarithmic (Log) contrasts.

**Table 4 molecules-27-04288-t004:** The effect on replacing diet with *Paulownia* leaves extract (PLE) on fatty acid (FA) proportion (mg/100 g of FA) in ruminal fluid.

Fatty Acid, mg/100 g FA	CON	PLE (mg/kg DM)						SEM	*p*-ANOVA	Contrast *			
		100	200	400	600	800	1000			L	Q	C	Log
C8:0	0.11	0.08	0.09	0.15	0.10	0.08	0.05	0.0067	0.348	0.134	0.392	0.773	0.045
C10:0	0.21	0.13	0.19	0.09	0.15	0.16	0.09	0.0154	0.457	0.41	0.195	0.109	0.032
C12:0	1.35	1.87	1.59	1.86	1.82	1.60	1.04	0.0582	0.316	0.376	0.282	0.300	0.021
C14:0	1.36	1.41	1.53	1.65	1.60	1.48	1.24	0.0293	0.764	0.618	0.700	0.901	0.099
C14:1	0.48	0.49	0.48	0.57	0.56	0.46	0.44	0.0184	0.196	0.375	0.184	0.080	0.046
C15:0	1.22	1.22	1.22	1.22	1.28	1.18	1.17	0.0246	0.901	0.195	0.101	0.054	0.001
C16:0	25.4	24.1	23.7	24.5	23.6	24.1	23.1	0.152	0.884	0.218	0.709	0.786	0.054
C16:1	0.46	0.44	0.43	0.44	0.45	0.40	0.39	0.0089	0.947	0.92	0.66	0.385	0.040
C17:0	0.66	0.76	0.71	0.65	0.67	0.66	0.60	0.0153	0.647	0.151	0.132	0.122	0.031
C18:0	29.4	28.8	29.4	27.7	26.1	25.3	24.1	0.318	0.273	0.712	0.221	0.078	<0.001
C18:1 trans-10	0.76	0.84	0.93	0.95	1.53	1.60	1.70	0.046	0.067	0.879	0.338	0.097	0.004
C18:1 cis-9	11.3	12.0	12.1	11.5	11.5	11.1	12.4	0.295	0.776	0.888	0.948	0.787	0.120
C18:1 cis-12	0.51	0.48	0.47	0.42	0.47	0.48	0.44	0.010	0.842	0.282	0.293	0.289	0.037
C18:2 cis-9, cis-12	8.25	8.28	8.22	10.0	11.3	12.5	14.6	0.308	0.050	0.97	0.375	0.148	0.047
C18:3 cis-9, cis-12, cis-15	0.87	1.10	1.21	1.24	1.55	1.68	2.05	0.082	0.026	0.589	0.899	0.554	0.049
C18:2 cis-9, trans-11	0.04	0.08	0.06	0.08	0.05	0.03	0.02	0.0033	0.881	0.557	0.545	0.572	0.032
C18:2 trans-10, cis-12	0.07	0.15	0.23	0.20	0.25	0.27	0.28	0.011	0.086	0.834	0.292	0.057	0.001
C18:3n-6	0.44	0.36	0.35	0.61	0.44	0.39	0.35	0.017	0.027	0.026	0.074	0.194	0.045
Sum of other FA ^#^	17.07	17.3	17.1	16.1	16.6	16.7	16.1	0.450	0.792	0.003	0.017	0.046	0.050
Sum of SFA	65.0	63.9	63.5	61.8	59.8	59.3	55.4	0.424	0.589	0.946	0.268	0.068	0.007
Sum of UFA	35.0	36.1	36.5	38.2	40.2	40.7	44.6	0.424	0.438	0.946	0.268	0.068	0.005
Sum of MUFA	25.1	26.0	26.3	25.9	26.5	25.7	27.3	0.245	0.805	0.876	0.498	0.279	0.019
Sum of PUFA	9.86	10.1	10.2	12.4	13.7	15.0	17.4	0.348	0.051	0.978	0.384	0.146	0.041
Sum of n-6 FA	9.20	9.13	9.04	11.1	12.2	13.4	15.4	0.303	0.067	0.842	0.299	0.114	0.044
Sum of n-3 FA	0.87	1.10	1.21	1.24	1.55	1.68	2.05	0.082	0.047	0.589	0.899	0.554	0.032
n-6/n-3 FA ratio	10.8	8.4	8.4	11.2	11.3	8.09	9.63	0.489	0.097	0.316	0.501	0.709	0.051
Sum of n-6 PUFA	8.69	8.65	8.57	10.6	11.7	12.9	14.9	0.304	0.128	0.871	0.318	0.124	0.008
Sum of n-3 PUFA	0.87	1.10	1.21	1.24	1.55	1.68	2.05	0.082	0.041	0.589	0.899	0.554	0.011
PUFA/SFA ratio	0.15	0.16	0.16	0.20	0.23	0.25	0.32	0.0071	0.052	0.879	0.432	0.156	0.009
LNA/LA ratio	0.11	0.14	0.15	0.13	0.15	0.14	0.14	0.0071	0.910	0.405	0.609	0.778	0.199
Sum of C18:1	20.9	22.1	22.4	21.9	22.7	22.3	23.8	0.284	0.785	0.792	0.349	0.149	0.002
Sum of trans C18:1	6.91	7.50	7.66	7.82	8.70	8.63	8.76	0.120	0.397	0.422	0.057	0.009	<0.001
Sum of cis C18:1	14.0	14.6	14.7	14.1	14.03	13.7	15.0	0.306	0.645	0.945	0.897	0.742	0.243
Sum of MCFA	36.7	36.1	34.8	35.1	34.6	34.6	32.1	0.251	0.368	0.617	0.778	0.375	0.021
Sum of LCFA	63.0	63.7	64.9	64.7	65.1	65.2	67.8	0.256	0.591	0.631	0.74	0.337	0.034
DI	0.167	0.179	0.178	0.174	0.179	0.173	0.196	0.0038	0.834	0.741	0.967	0.726	0.398
DI C14:1 cis- 9/(C14:0+C14:1 cis-9)	0.26	0.26	0.24	0.25	0.26	0.24	0.26	0.006	0.939	0.315	0.173	0.101	0.002
DI C16:1 cis-9/(C16:0+C16:1 cis-9)	0.018	0.018	0.018	0.018	0.019	0.017	0.017	0.00034	0.991	0.845	0.631	0.472	0.019
DI C18:1 cis-9/(C18:0+C18:1 cis-9)	0.27	0.29	0.29	0.29	0.30	0.30	0.34	0.0064	0.846	0.879	0.718	0.452	0.044
DI RA/(VA+RA)	0.009	0.014	0.012	0.014	0.008	0.005	0.004	0.0006	0.730	0.562	0.702	0.83	0.656

SFA, saturated fatty acids; UFA, unsaturated fatty acids; MUFA, monounsaturated fatty acids; PUFA, polyunsaturated fatty acids; LNA, linolenic acid; LA, linoleic acid; MCFA, medium-chain fatty acids; LCFA, long-chain fatty acids; RA, rumenic acid; VA, vaccenic acid; DI, desaturation index. * Contrast: linear (L), quadratic (Q), cubic (C), and logarithmic (Log) contrasts; ^#^ Sum of other FA: C13:0, C:14:1 iso, C14:1 aiso, C15:1, C15:1 aiso, C16:1 iso, C17:1 aiso, C17:1, C17:0 iso, C18:1 trans-6-8, C18:1 trans-9, C18:1 trans-11, C18:1 cis-11, C18:1 cis-13, C18:1 cis-14, C18:2 trans-11 cis-15, C20:0, C20:1-trans, C21:0, C18:3 cis- 9 trans-11 cis-15, C22:0, C23:0, C24:0, and C24:1.

**Table 5 molecules-27-04288-t005:** Forward and reverse primers used in ruminal bacteria RT-PCR analysis.

Target	Primer Sequence (5′ to 3′)	Reference
*Ruminococcus flavefaciens*	F: CGAACGGAGATAATTTGAGTTTACTTAGG	[48]
	R: CGGTCTCTGTATGTTATGAGGTATTACC	
*Fibrobacter succinogenes*	F: GTTCGGAATTACTGGGCGTAAA	[49]
	R: CGCCTGCCCCTGAACTATC	
*Streptococcus bovis*	F: TTCCTAGAGATAGGAAGTTTCTTCGG	[50]
	R: ATGATGGCAACTAACAATAGGGGT	
*Prevotella* spp.	F: GAAGGTCCCCCACATTG	[50]
	R: CAATCGGAGTTCTTCGTG	
*Butyrivibrio proteoclasticus*	F: TCCTAGTGTAGCGGTGAAATG	[51]
	R: TTAGCGACGGCACTGAATGCCTA	
*Ruminococcus albus*	F: CCCTAAAAGCAGTCTTAGTTCG	[52]
	R CCTCCTTGCGGTTAGAACA	
*Butyrivibrio fibrisolvens*	F: ACACACCGCCCGTCACA	[53]
	R: TCCTTACGGTTGGGTCACAGA	
*Anaerovibrio lipolytica*	F: GAAATGGATTCTAGTGGCAAACG	[54]
	R: ACATCGGTCATGCGACCAA	
*Megasphaera elsdenii*	F: AGATGGGGACAACAGCTGGA	[50]
	R: CGAAAGCTCCGAAGAGCCT	

## Data Availability

The data presented in this study are available on reasonable request from the corresponding author.

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
