# Peer review of "Effect of Paulownia Leaves Extract Levels on In Vitro Ruminal Fermentation, Microbial Population, Methane Production, and Fatty Acid Biohydrogenation"

_molecules, 2022, doi:10.3390/molecules27134288_

Round 1

Reviewer 1 Report

The manuscript provides new information about a novel broswe plant whoose leaves can be fed to ruminants as an anti-methanogenic agent. Overall, the manuscript was well written and the study executed with careful and well layout plan. 

However, the manuscript should include line numbering for easier review. The overall study objective is well stated. The methodology employed is repeatable and conventional.

 Secific comments include: 

First paragraph/1st sentence need to be re-casted. The cause effect relationship is not clear.

Paulownia dried leaves affected rumen fermentation? How? Authors must state the type of influence? Gas production? Increased or decreased?

Puchalska et al [xx] use appropriate reference style.

Results

The results of VFA should be expressed in standard conventional forms as molar proportion. Not mM.

The conlusion is very lucid. 

Materials and Method

What is the difference between control group and ethanol group? Ethanol inclusion in blank? Why was this included ?How was the blank treated subsequently?

Was there any gas sampling? This is not properly described. The information that gas was sampled at 24 h is not sufficient. Was there a one time sampling? This is not mentioned. If gas was not sampled, before the 24 h mark, there will exist a confounding effect of trapped gas on organic matter digestibility.

Statistical analysis: The ANOVA output should generate a p- value indicating the significance of treatment means based on the model statement. This should have been included aside the p-values of the contrast statements.

Author Response

Poznan, June 28th, 2022

Editorial Office

Molecules

Guest Editors

Prof. Dr. Rudolf Bauer

Dr. Jelena S. Katanic Stankovic

Dear Editors,

Manuscript Number:  molecules-1802934

Dear Editors and Reviewers, 

We would like to thank you for all of your comments and corrections.

We arranged our changes as follows:

- changed words/added or properly ordered were highlighted in yellow 

-AU: stands for our replies, answers and comments

Below, please find our responses, point-by-point comments and suggestions.

Reviewer 1

The manuscript provides new information about a novel broswe plant whoose leaves can be fed to ruminants as an anti-methanogenic agent. Overall, the manuscript was well written and the study executed with careful and well layout plan. 

However, the manuscript should include line numbering for easier review. The overall study objective is well stated. The methodology employed is repeatable and conventional.

 Secific comments include: 

First paragraph/1st sentence need to be re-casted. The cause effect relationship is not clear.

AU: Thank you for the suggestion. We re-casted the first paragraph hoping that it now makes sense clearly.

Growing consumer awareness [1], the need to feed a growing population, and protect the natural environment [2] are the driving force behind the search for alternative feed components that do not compete with human food and are an ally in climate protection. These include alternative dietary components used in cattle nutrition [3] and waste management. Ecologists focus on implementing those solutions in high-tech and urbanized countries, where such adverse impacts are evident [1].

Paulownia dried leaves affected rumen fermentation? How? Authors must state the type of influence? Gas production? Increased or decreased? 

AU: Thank you for your comment. We improved the sentence as follows:

Paulownia dried leaves or silage positively affected the rumen fermentation parameters by changing acetate to propionate ratio without any adverse effect on nutrient digestion and, therefore, exploited in the nutrition of ruminants [3,9].

Puchalska et al [xx] use appropriate reference style.

AU: The appropriate style was used.

Results

The results of VFA should be expressed in standard conventional forms as molar proportion. Not mM.

AU: Thank you for your comment. We changed it.

The conlusion is very lucid. 

Materials and Method

What is the difference between control group and ethanol group? Ethanol inclusion in blank? Why was this included ?How was the blank treated subsequently? 

Was there any gas sampling? This is not properly described. The information that gas was sampled at 24 h is not sufficient. Was there a one time sampling? This is not mentioned. If gas was not sampled, before the 24 h mark, there will exist a confounding effect of trapped gas on organic matter digestibility.

AU: The explanation is as follows:

We had to dissolve PLE in ethanol to be able to apply different amounts of the extract precisely, so we made an additional group where we used ethanol in the same amount as in the groups with the addition of the extract. After the statistical analysis of the obtained results, between the control group and the group with ethanol, we found that ethanol had no effect in the amounts used on the parameters tested, therefore we excluded it from the table. Blank samples are used in our laboratory as a routine procedure to see what the buffered ruminal fluid itself produces - mostly gas. The values ​​related to, among others, gas production or methane concentration are subtracted from the results obtained in the actual experiment in order to be able to verify the changes of the examined indicators on the basis of the substrate and experimental factor used only.

The batch culture system used does not require additional gas collection during the 24 h incubation procedure. The 400 mg of the substrate suspended in 40 ml of buffered rumen fluid in a 24-hour incubation did not produce an excessive amount of gasses. This procedure is commonly used in many laboratories, and incubation is most often completed after 24 hours.

Statistical analysis: The ANOVA output should generate a p- value indicating the significance of treatment means based on the model statement. This should have been included aside the p-values of the contrast statements.

AU: We added p-values from analyses of variance in Tables 3, 4, and 5.

Reviewer 2

The manuscript entitled “Investigation of Paulownia leaves on in vitro ruminal fermentation, microbial population, methane production, and fatty acid biohydrogenation” reports the results of a study that assess the potential of Paulownia leaves as potential feed additive for ruminants. This manuscript reports on a topic pertinent to ruminant nutrition. The reliable knowledge of the optimal doses of the newly natural feed additives it is of the utmost importance for ruminant feeding systems. The manuscript is well written and organized. The data presented are good and sufficient to provide very good quality information to contrast the hypothesis raised. Material and methods are exhaustively described and discussion is well supported. In my opinion, only few little flaws which should be rectified before publication. The observations and corrections must be performed are indicated in the reviewed manuscript.

AU: Thank you for all comments that allowed us to improve the manuscript.

I think the more adequate title is (hint): Paulownia leaves extract dosage levels on in vitro ruminal fermentation, microbial population, methane production, and fatty acid biohydrogenation

AU: Thank you for the hint. Taking it under consideration we suggest the following title:

Effect of Paulownia leaves extract levels on in vitro ruminal fermentation, microbial population, methane production, and fatty acid biohydrogenation

In text is expressed as mg/ g DM, but in Tables is expressed as mg/kg DM. Please correct this.

AU: We improved the units in table 1. It has to be mg/g DM. Thank you for your suggestion.

A tendency? or "Decrease in pH value (linear component, P=0.05) was observed with..." Please, define in statistical analyses when the contrast are considered significant

AU: We corrected this sentence. The new text is as follows: „Decrease in pH (linear, P=0.05) was observed with…”.

This statement is incongruent. If effect on pH was detected as "linear component" is incorrect mention "being maximals " at intermediate doses. This is only correct with quadratic effects. Please correct this

AU: We corrected this sentence. The new text is: “with the highest pH at 100 mg/kg of PLE (Table 2).”.

How do you explain (in biological process) a cubic response in a dose-dependent experiment. For example, according to propionate values, is a linear response, even thought the statistical analyses shown significance for all components (the same for total gas, CH4, and others variables).  On the other hand, Butyrivibrio population was in a quadratic manner, but all components were statistically significant.

AU: From a statistical point of view: in each of the cases, the fit to the model was very high and was characterized by high values ​​of the coefficient of determination. Dear Reviewer, if the explanation is not clear enough, we would be grateful for any hint. Thank you in advance.

Really? Even when the values of propionate concentration are are steadily increasing (and SEM= 0.8% of the mean value)  You detect a linear, quadratic and cubic significance? Please, perform confidence interval analysis at 95% between the values here and in all values in Tables). Orthogonal polynomials were analyzed as unequally spaced???

AU: Yes, all: linear, quadratic, cubic and logarithmic were statistically significant for propionate. Coefficients of determination were equal to 89.6%, 73.3%, 59.3% and 89.6%, respectively.

Due to the very complex tables (presentation of contrasts), the confidence intervals will not add much to the presentation of the results. Therefore, we would like to ask the Reviewer do not to include them in the table. If the Reviewer still believes it is required, we will include this information in the table.

A tendency? or "Decrease in pH value (linear component, P=0.05) was observed with..." Please, define in statistical analyses when the contrast are considered significant

AU: We corrected this sentence. We added in “6. Statistical analysis“ section: “Contrast are considered significant when the p-value ≤0.05, and tendencies are identified when the p-value >0.05 to ≤0.10.”

orthogonal polynomials were analyzed as unequally spaced??? Please specified

AU: Orthogonal polynomials were used in the analysis of variance for the construction of contrasts among unequally spaced levels of a treatment factor.

Please, indicate when the contrasts are considered significant (i.e., contrast are considered significant when the P value ≤ 0.05, and tendencies are identified when the P -value > 0.05 and ≤ 0.10).

AU: We corrected this sentence. We added in “6. Statistical analysis“ section: “Contrast is considered significant when the p-value ≤0.05, and tendencies are identified when the p-value >0.05 to ≤0.10.”

Please rewrite as: The greatest mitigation was noted with the highest dose of PLE in buffered ruminal fluid (equivalent to 8.4 g phenolic compounds/L of ruminal fluid). This mitigation was directly related to a linear decrease of the degradability of feed (substrate).

AU: Thank you for your comment. The sentence was rewritten.

We would like once again to thank the Editors and Reviewers very much for all the valuable comments and suggestions that helped us to improve our manuscript.

Yours sincerely

Adam Cieślak

Reviewer 2 Report

The manuscript entitled “Investigation of Paulownia leaves on in vitro ruminal fermentation, microbial population, methane production, and fatty acid biohydrogenation” reports the results of a study that assess the potential of Paulownia leaves as potential feed additive for ruminants. This manuscript reports on a topic pertinent to ruminant nutrition. The reliable knowledge of the optimal doses of the newly natural feed additives it is of the utmost importance for ruminant feeding systems. The manuscript is well written and organized. The data presented are good and sufficient to provide very good quality information to contrast the hypothesis raised. Material and methods are exhaustively described and discussion is well supported. In my opinion, only few little flaws which should be rectified before publication. The observations and corrections must be performed are indicated in the reviewed manuscript.

Author Response

Poznan, June 28th, 2022

Editorial Office

Molecules

Guest Editors

Prof. Dr. Rudolf Bauer

Dr. Jelena S. Katanic Stankovic

Dear Editors,

Manuscript Number:  molecules-1802934

Dear Editors and Reviewers, 

We would like to thank you for all of your comments and corrections.

We arranged our changes as follows:

- changed words/added or properly ordered were highlighted in yellow 

-AU: stands for our replies, answers and comments

Below, please find our responses, point-by-point comments and suggestions.

Reviewer 2

The manuscript entitled “Investigation of Paulownia leaves on in vitro ruminal fermentation, microbial population, methane production, and fatty acid biohydrogenation” reports the results of a study that assess the potential of Paulownia leaves as potential feed additive for ruminants. This manuscript reports on a topic pertinent to ruminant nutrition. The reliable knowledge of the optimal doses of the newly natural feed additives it is of the utmost importance for ruminant feeding systems. The manuscript is well written and organized. The data presented are good and sufficient to provide very good quality information to contrast the hypothesis raised. Material and methods are exhaustively described and discussion is well supported. In my opinion, only few little flaws which should be rectified before publication. The observations and corrections must be performed are indicated in the reviewed manuscript.

AU: Thank you for all comments that allowed us to improve the manuscript.

I think the more adequate title is (hint): Paulownia leaves extract dosage levels on in vitro ruminal fermentation, microbial population, methane production, and fatty acid biohydrogenation

AU: Thank you for the hint. Taking it under consideration we suggest the following title:

Effect of Paulownia leaves extract levels on in vitro ruminal fermentation, microbial population, methane production, and fatty acid biohydrogenation

In text is expressed as mg/ g DM, but in Tables is expressed as mg/kg DM. Please correct this.

AU: We improved the units in table 1. It has to be mg/g DM. Thank you for your suggestion.

A tendency? or "Decrease in pH value (linear component, P=0.05) was observed with..." Please, define in statistical analyses when the contrast are considered significant

AU: We corrected this sentence. The new text is as follows: „Decrease in pH (linear, P=0.05) was observed with…”.

This statement is incongruent. If effect on pH was detected as "linear component" is incorrect mention "being maximals " at intermediate doses. This is only correct with quadratic effects. Please correct this

AU: We corrected this sentence. The new text is: “with the highest pH at 100 mg/kg of PLE (Table 2).”.

How do you explain (in biological process) a cubic response in a dose-dependent experiment. For example, according to propionate values, is a linear response, even thought the statistical analyses shown significance for all components (the same for total gas, CH4, and others variables).  On the other hand, Butyrivibrio population was in a quadratic manner, but all components were statistically significant.

AU: From a statistical point of view: in each of the cases, the fit to the model was very high and was characterized by high values ​​of the coefficient of determination. Dear Reviewer, if the explanation is not clear enough, we would be grateful for any hint. Thank you in advance.

Really? Even when the values of propionate concentration are are steadily increasing (and SEM= 0.8% of the mean value)  You detect a linear, quadratic and cubic significance? Please, perform confidence interval analysis at 95% between the values here and in all values in Tables). Orthogonal polynomials were analyzed as unequally spaced???

AU: Yes, all: linear, quadratic, cubic and logarithmic were statistically significant for propionate. Coefficients of determination were equal to 89.6%, 73.3%, 59.3% and 89.6%, respectively.

Due to the very complex tables (presentation of contrasts), the confidence intervals will not add much to the presentation of the results. Therefore, we would like to ask the Reviewer do not to include them in the table. If the Reviewer still believes it is required, we will include this information in the table.

A tendency? or "Decrease in pH value (linear component, P=0.05) was observed with..." Please, define in statistical analyses when the contrast are considered significant

AU: We corrected this sentence. We added in “6. Statistical analysis“ section: “Contrast are considered significant when the p-value ≤0.05, and tendencies are identified when the p-value >0.05 to ≤0.10.”

orthogonal polynomials were analyzed as unequally spaced??? Please specified

AU: Orthogonal polynomials were used in the analysis of variance for the construction of contrasts among unequally spaced levels of a treatment factor.

Please, indicate when the contrasts are considered significant (i.e., contrast are considered significant when the P value ≤ 0.05, and tendencies are identified when the P -value > 0.05 and ≤ 0.10).

AU: We corrected this sentence. We added in “6. Statistical analysis“ section: “Contrast is considered significant when the p-value ≤0.05, and tendencies are identified when the p-value >0.05 to ≤0.10.”

Please rewrite as: The greatest mitigation was noted with the highest dose of PLE in buffered ruminal fluid (equivalent to 8.4 g phenolic compounds/L of ruminal fluid). This mitigation was directly related to a linear decrease of the degradability of feed (substrate).

AU: Thank you for your comment. The sentence was rewritten.

Reviewer 1

The manuscript provides new information about a novel broswe plant whoose leaves can be fed to ruminants as an anti-methanogenic agent. Overall, the manuscript was well written and the study executed with careful and well layout plan. 

However, the manuscript should include line numbering for easier review. The overall study objective is well stated. The methodology employed is repeatable and conventional.

 Secific comments include: 

First paragraph/1st sentence need to be re-casted. The cause effect relationship is not clear.

AU: Thank you for the suggestion. We re-casted the first paragraph hoping that it now makes sense clearly.

Growing consumer awareness [1], the need to feed a growing population, and protect the natural environment [2] are the driving force behind the search for alternative feed components that do not compete with human food and are an ally in climate protection. These include alternative dietary components used in cattle nutrition [3] and waste management. Ecologists focus on implementing those solutions in high-tech and urbanized countries, where such adverse impacts are evident [1].

Paulownia dried leaves affected rumen fermentation? How? Authors must state the type of influence? Gas production? Increased or decreased? 

AU: Thank you for your comment. We improved the sentence as follows:

Paulownia dried leaves or silage positively affected the rumen fermentation parameters by changing acetate to propionate ratio without any adverse effect on nutrient digestion and, therefore, exploited in the nutrition of ruminants [3,9].

Puchalska et al [xx] use appropriate reference style.

AU: The appropriate style was used.

Results

The results of VFA should be expressed in standard conventional forms as molar proportion. Not mM.

AU: Thank you for your comment. We changed it.

The conlusion is very lucid. 

Materials and Method

What is the difference between control group and ethanol group? Ethanol inclusion in blank? Why was this included ?How was the blank treated subsequently? 

Was there any gas sampling? This is not properly described. The information that gas was sampled at 24 h is not sufficient. Was there a one time sampling? This is not mentioned. If gas was not sampled, before the 24 h mark, there will exist a confounding effect of trapped gas on organic matter digestibility.

AU: The explanation is as follows:

We had to dissolve PLE in ethanol to be able to apply different amounts of the extract precisely, so we made an additional group where we used ethanol in the same amount as in the groups with the addition of the extract. After the statistical analysis of the obtained results, between the control group and the group with ethanol, we found that ethanol had no effect in the amounts used on the parameters tested, therefore we excluded it from the table. Blank samples are used in our laboratory as a routine procedure to see what the buffered ruminal fluid itself produces - mostly gas. The values ​​related to, among others, gas production or methane concentration are subtracted from the results obtained in the actual experiment in order to be able to verify the changes of the examined indicators on the basis of the substrate and experimental factor used only.

The batch culture system used does not require additional gas collection during the 24 h incubation procedure. The 400 mg of the substrate suspended in 40 ml of buffered rumen fluid in a 24-hour incubation did not produce an excessive amount of gasses. This procedure is commonly used in many laboratories, and incubation is most often completed after 24 hours.

Statistical analysis: The ANOVA output should generate a p- value indicating the significance of treatment means based on the model statement. This should have been included aside the p-values of the contrast statements.

AU: We added p-values from analyses of variance in Tables 3, 4, and 5.

We would like once again to thank the Editors and Reviewers very much for all the valuable comments and suggestions that helped us to improve our manuscript.

Yours sincerely

Adam Cieślak

Round 2

Reviewer 1 Report

The authors have adopted the recommendations from the previous review. Only one item is missing. The authors provided a clearer response to the initial question in Section 4.2. However, it will make for easy replication of the study if this vital information is added to the paper.

e.g. y gramm of PLE was dissolved in ethanol x mL ethanol and added to the substrate at xy mL. 

Was the PLE solution added a day to the in vitro incubation to allow evaporation or added on the same day of incubation? While it is acceptable that the ethanol effect may be insignificant, it can have a confounding effect. It is important to document the procedure and let others in the future adopt it or seek a better way of doing it.

Author Response

Manuscript Number:  molecules-1802934

Dear Reviewer, 

We would like to thank you for all of your comments.

We arranged our changes as follows:

- changed words/added or properly ordered were highlighted in yellow 

-AU: stands for our replies, answers and comments

Below, please find our responses.

Reviewer 2

The authors have adopted the recommendations from the previous review. Only one item is missing. The authors provided a clearer response to the initial question in Section 4.2. However, it will make for easy replication of the study if this vital information is added to the paper.

e.g. y gramm of PLE was dissolved in ethanol x mL ethanol and added to the substrate at xy mL. 

Was the PLE solution added a day to the in vitro incubation to allow evaporation or added on the same day of incubation? While it is acceptable that the ethanol effect may be insignificant, it can have a confounding effect. It is important to document the procedure and let others in the future adopt it or seek a better way of doing it.

AU: Thank you very much for your comments. Necessary improvement was introduced into the manuscript. Now it is as follows:

All six doses (representing 4, 8, 16, 24, 32, and 40 mg per bottle) were dissolved in 100 µL of ethanol and added to the substrate 2 h before the start of filling the bottle with buffered rumen fluid. A small amount of ethanol addition 2 h before the start of the experiment allowed evaporation of ethanol from the substrate.

We would like once again to thank the Editors and Reviewer very much for all the valuable comments and suggestions that helped us to improve our manuscript.

Yours sincerely

Adam Cieślak